# Multi-Layered Regulations on the Chromatin Architectures: Establishing the Tight and Specific Responses of Fission Yeast *fbp1* Gene Transcription

**DOI:** 10.3390/biom12111642

**Published:** 2022-11-05

**Authors:** Ryuta Asada, Kouji Hirota

**Affiliations:** 1Department of Viticulture and Enology, University of California, Davis, CA 95616, USA; 2Department of Chemistry, Graduate School of Science, Tokyo Metropolitan University, Hachioji 192-0397, Tokyo, Japan

**Keywords:** chromatin, histone, nucleosome, transcription factor, non-coding RNA (ncRNA), metabolic stress response, *S. pombe*

## Abstract

Transcriptional regulation is pivotal for all living organisms and is required for adequate response to environmental fluctuations and intercellular signaling molecules. For precise regulation of transcription, cells have evolved regulatory systems on the genome architecture, including the chromosome higher-order structure (e.g., chromatin loops), location of transcription factor (TF)-binding sequences, non-coding RNA (ncRNA) transcription, chromatin configuration (e.g., nucleosome positioning and histone modifications), and the topological state of the DNA double helix. To understand how these genome-chromatin architectures and their regulators establish tight and specific responses at the transcription stage, the fission yeast *fbp1* gene has been analyzed as a model system for decades. The fission yeast *fbp1* gene is tightly repressed in the presence of glucose, and this gene is induced by over three orders of magnitude upon glucose starvation with a cascade of multi-layered regulations on various levels of genome and chromatin architecture. In this review article, we summarize the multi-layered transcriptional regulatory systems revealed by the analysis of the fission yeast *fbp1* gene as a model system.

## 1. Introduction

Transcriptional control is essential for all organisms to adapt to environmental changes and respond adequately to intercellular signals. Abnormalities in transcriptional regulation are associated with serious diseases, including cancer [1]. Thus, exploring transcriptional regulatory systems is an important research subject. The response of the proper gene(s) to the proper timing is accomplished by the regulation of transcription factor (TF) binding to specific target DNA sequences. Numerous TFs are encoded in the genome, and their binding profiles determine the transcriptomes, such as genes that are activated or repressed and the timing of the same [2,3]. Generally, the genome architecture, including the chromatin structure and the location of the TF binding sequence in the chromatin environment, plays a critical role in determining TF binding profiles as the access of regulatory proteins such as TFs to their target DNAs is restricted by the tightly positioned nucleosomes formed on the genomic DNA [4]. Thus, the tight regulation of chromatin configurations, open (nucleosome-less) and closed (nucleosome-condensed), around such DNA elements results in tight switching of the transcription state between active transcription and gene silencing [5,6]. Chromatin is highly three-dimensionally organized in the limited space of the nucleus. The three-dimensional positioning or regulation of the proximity of DNA elements in three-dimensional space provides further complexity for transcriptional regulation by organizing the timing of transcription regulator recruitment, as represented by enhancer-promoter contact [7].

These multiple layers of transcriptional regulatory networks in the genome architecture are regulated by many factors, including histone modification enzymes, ATP-dependent chromatin remodelers, and non-coding RNAs. The N-terminal tail of histones, components of nucleosomes, are frequently post-translationally modified, and the marks are generally linked to specific chromatin states (active or repressive) [8]. Acetylation is an example of a histone mark for active chromatin. Histone acetyltransferases (HATs) add acetyl groups, and the acetylated histone tails are recognized by ATP-dependent chromatin remodelers, which slide or disassemble nucleosomes, resulting in the generation of open chromatin suitable for active transcription [9,10,11]. Small and long non-coding RNAs are also involved in the regulation of the chromatin genome architecture. Recent advances in comprehensive transcriptome sequencing have revealed that the majority of transcripts do not originate from protein-coding genes in human or higher organism genomes, and non-coding RNAs transcribed from intergenic and antisense regions are highlighted as important regulators of gene regulation, particularly in the transcription process [12,13]. Small non-coding RNAs, such as siRNA and piRNA, and their processing machinery target nascent transcripts and recruit heterochromatin machinery for the silencing of repeated or transposable genomic elements [14,15]. Myriad long non-coding RNAs (lncRNAs) are involved in the modulation of multiple-layered genome architectures in both transcription repression and activation processes. These include lncRNA molecules that directly interact with chromatin modifiers or transcription machinery and mediate the recruitment of these regulators to specific gene loci or facilitate the formation of a three-dimensional chromatin loop structure for transcription activation [16,17,18]. 

Although many studies have focused on each regulatory layer and regulation factor in many model organisms, there is a lack of clarity on how these multiple layered regulations manipulated by many factors on the genome architecture are coordinated to establish tight and specific regulation of transcription. Fission yeast *fbp1* is a gene in which transcription is tightly regulated in response to environmental glucose concentration, and a series of studies over several decades have revealed the mechanism to establish glucose starvation stress-specific regulation of *fbp1* transcription, which is complexly regulated by the coordination of multi-layered regulations at local chromatin-genome architectures including chromatin configurations, three-dimensional chromatin loop, local inter-TF direct interaction, and lncRNA regulation. In this review, we summarize the multi-layered regulation of *fbp1* transcription and present a model for the tight and specific regulation of transcription.

## 2. Signal Pathways and Transcription Factors Required for the Regulation of *fbp1* Gene

In the fission yeast *Schizosaccharomyces pombe*, the *fbp1* gene, which encodes fructose-1,6-bisphosphatase (FBPase), an essential enzyme for gluconeogenesis, displays dramatic transcriptional regulation and is regulated several hundred-fold by glucose concentration [19]. This enzyme is conserved from yeasts to humans. Mammalian FBPase activity is regulated by two metabolic inhibitors, AMP and fructose-2,6-bisphosphate [20], whereas FBPase activity in the budding yeast *Saccharomyces cerevisiae* is controlled by both glucose-dependent repression of gene transcription and glucose inactivation of the enzyme itself [21,22]. On the other hand, the FBPase activity in *S. pombe* is regulated only by transcriptional control but not by direct glucose inactivation of the enzyme [19], which is reminiscent of the requirement to establish systems for a quick but highly specific transcriptional response to the environmental glucose concentration. 

Studies exploring the signaling pathways and transcriptional regulation systems for the glucose-dependent control of *fbp1* transcription in *S. pombe* were initiated by Dr. Charles S. Hoffman. He generated a strain carrying the *fbp1-ura4* fusion gene, which is phenotypically uracil auxotrophic (Ura−) under glucose-rich conditions. He screened uracil prototroph (Ura+) colonies and identified several glucose-insensitive transcription (*git*) mutants [23,24,25,26,27,28,29,30]. His *git*-gene screening and contemporaneous studies on fission yeast sexual development by Dr. Masayuki Yamamoto determined the framework of the protein kinase A (PKA) pathway [23,24,25,26,27,28,29,30,31,32,33,34] (Figure 1A). Glucose signaling is mediated through the seven-transmembrane receptor coupled with trimeric G proteins and triggers the activation of adenylate cyclase, which in turn activates PKA by the inhibition of the regulatory subunit Cgs1 by cAMP to repress *fbp1* (Figure 1A). PKA activity represses a C_2_H_2_ zinc-finger transcription factor, Rst2, by sequestering it in the cytoplasm, and the repression of PKA activity by glucose starvation induces rapid Rst2 nuclear import for the transcription activation of *fbp1* and other genes [32,35,36] (Figure 1A,B). Glucose starvation also stimulates the stress-activated mitogen-activated protein kinase (MAPK) pathway (Figure 1B). Atf1, a basic leucine zipper (bZIP) transcription factor activated by the MAPK pathway, is pivotal for *fbp1* induction upon glucose starvation [37,38,39,40,41]. The initial study of comprehensive deletion of the segments at the *fbp1* promoter region identified two regulatory elements, upstream activation sequences 1 and 2 (UAS1 and UAS2) [39]. The identified UAS1 sequence is consistent with a well-conserved Atf1 binding site called cAMP response element (CRE) [37,39,42], whereas UAS2 has a CT-rich stress response element (STRE; CCCCTC), which serves as an Rst2 binding site in the *ste11* gene promoter region [32] (Figure 1C). It is known that the STRE sequence is also targeted by another Zn finger transcription factor, Scr1, and analysis of the deletion mutant identified the Scr1 role in *fbp1* transcription repression [36,39] (Figure 1C). In contrast to Rst2, Scr1 is localized to the nucleus under glucose-rich conditions and is rapidly transported to the cytoplasm during glucose starvation [36]. The reciprocal binding of Rst2 and Scr1 to UAS2 creates tight on/off *fbp1* transcription [36]. The essential event of Rst2 binding to UAS2 for *fbp1* transcriptional activation is not just a simple regulation of Rst2 nuclear and cytoplasmic localization. Additional segment deletion research around UAS1 found another Rst2 binding CT-rich sequence [43] (Figure 1C). After induction of glucose starvation, Rst2 initially binds to the site near UAS1 and is further delivered to UAS2 via a local chromatin loop (see Section 6.2) [43]. In addition, multicopy suppressor screening of the *cgs1* mutant, which fails to induce *fbp1* transcription due to high PKA activity, identified another TF, CBF (CCAAT-binding factor), and global Tup family co-repressor [44]. CBF is a well conserved heterotrimeric TF composed by Php2, Php3, and Php5 and the mammalian homolog, NF-Y, is often detected near the promoter to generate open chromatin around transcription start site [45,46,47]. Although the exact binding site at the *fbp1* promoter has not yet been identified, it is estimated by chromatin immunoprecipitation and is close to UAS2 [48] (Figure 1C). The Tup family co-repressor is a conserved groucho-TLE-type transcriptional repressor. *S. cerevisiae* Tup1 is a well-studied homolog, and Tup1 is recruited to the gene promoter region via interaction with sequence-specific DNA-binding proteins [49]. Tup1 has multiple ways to repress gene transcription, recruitment of histone deacetylase complexes, modulation of the nucleosome positioning to mask TF binding sites, and inhibition of the transcription machinery loading to the promoter [49]. Although Tup1 represses the transcription of many genes, some studies have shown that this is not just a repressor, and in some cases, Tup1 is required for transcription activation, suggesting that it seems to have a more important function in the proper regulation of transcription on and off switching [50,51]. *fbp1* studies provided mechanistic insights into Tup co-repressor functions to generate highly stress-specific induction of transcription. Fission yeast encodes two Tup1 co-repressor genes, *tup11* and *tup12*, and double deletion causes impaired glucose starvation-specific induction of *fbp1* mRNA, i.e., the *fbp1* gene can be induced under other stresses (stationary phase, osmotic stress, and nitrogen starvation) [52]. Subsequent studies found that Tup11 and Tup12 work to build multiple phases of repression in the process of *fbp1* gene induction, including inhibition of open chromatin formation, stable TF binding to the target sites, and recruitment of transcription machinery to the *fbp1* promoter, all of which are antagonized by many factors and mechanisms such as TFs, lncRNAs, and three-dimensional genome architecture (detailed in the following sections). This multi-step antagonistic regulation against multiple Tup11/12 mediated repressions potentially generates stress specificity as the *fbp1* gene can be induced only when all antagonizing factors are activated. 

## 3. Chromatin Dynamics at *fbp1* Promoter Region

The importance of chromatin structure in transcriptional regulation had already been highlighted [5,6]. However, the chromatin state at *fbp1* gene regulatory regions had not yet been known. Indirect end-labeling analysis using micrococcal-nuclease (MNase)-digested chromatin was employed to reveal nuclease-hypersensitive sites, which reflect an open chromatin configuration. This revealed that chromatin-DNA around UAS1 and UAS2 was protected from MNase digestion in glucose-rich conditions, while MNase hypersensitive bands appeared at UAS1 and the region between UAS2 and TATA box 3 h after glucose starvation when *fbp1* was fully expressed [53]. These results indicate that the condensed chromatin configuration at UAS1 and UAS2 in repressive conditions is converted into an open state during *fbp1* transcriptional activation. The chromatin remodeling kinetics were analyzed further in the shorter time points (10, 20, 30, and 60 min) after glucose starvation. It was observed that chromatin at the *fbp1* upstream region was progressively converted into an open configuration, and it was induced by the several species of lncRNAs transcription through the *fbp1* upstream region [54] (Figure 2 and Figure 3). While the longest lncRNA (here termed as –a) was weakly transcribed in glucose rich repressive condition, at the initial time point after glucose starvation (10 min), the cascade transcriptions of lncRNA species (–b and c) were initiated following the open chromatin formation at UAS1. In the following time points, 20–30 min after glucose starvation, chromatin between UAS1 and UAS2 converted into open configuration, then, 60 min later, chromatin at TATA-box becomes open and the massive induction of *fbp1*–mRNA occurred [54]. Insertion of a transcription terminator sequence into the *fbp1* upstream region abolished both the cascade of lncRNA transcription and progressive chromatin alteration [54], showing the critical role of lncRNA transcriptions through the promoter region in chromatin relaxation to make accessible DNA sequences for the targeting of transcriptional activators and RNAPII (Section 4). These lncRNAs involved in the chromatin modulation were initially defined as ‘mRNA type long ncRNAs’ or ‘mlonRNAs’, when the term ‘lncRNAs’ had not been well recognized [55,56]. However, after this definition, the term ‘lncRNA’ has been commonly used for ‘mRNA-type long ncRNA’, and thus the definition of mlonRNA was changed to indicate ‘metabolic stress-induced lncRNAs’ [57] with the identification of mlonRNA type lncRNAs in genome-wide RNA-seq analysis [58]. 

In addition to mlonRNA-transcription-mediated chromatin remodeling, the analysis of the three identified TF (Atf1, Php5, and Rst2)-deletion mutants revealed that the chromatin at *fbp1* upstream region was regulated in three steps separating the region, around the UAS1, UAS1–UAS2 region, and around the TATA box [48] (Figure 2). Atf1 deletion fully inhibits the chromatin remodeling reaction at the *fbp1* upstream region including an initial point around UAS1 and its downstream region, indicating that Atf1 is required for the initiation of a series of chromatin alterations. Atf1 binds to UAS1, which causes chromatin alteration at UAS1, and also induces a cascade of mlonRNAs, resulting in the induction of chromatin remodeling in the region between UAS1 and UAS2 (Figure 2b,c). By generating open chromatin around UAS2, CBF (Php5) can be loaded to the target site, which induces chromatin opening at the TATA box located close downstream of the CBF binding site (Figure 2d). Thus, the chromatin upstream of *fbp1* is fully active, but the transcription of the *fbp1*-mRNA from TATA-box still requires Rst2. The analysis of Rst2 deletion revealed that Rst2 only affects the chromatin around UAS1, and the majority of the chromatin remodeling dynamics is not affected, though it is required for the recruitment of transcription machinery to the open chromatin TATA box. For this Rst2 function, Rst2 binding to UAS2 is required, and this is mediated by the formation of local chromatin loop structure [43]. Rst2 initially binds to the target site near UAS1 (CT-rich motif), and Rst2 is delivered to UAS2 associated with chromatin loop formation (Figure 2e,f) (Section 6.2). More importantly, the TFs requirement for chromatin and transcription machinery recruitment is canceled by deletion of Tup11/12 [48] (described in Section 6). This indicates that the Tup co-repressors generate multiple phases of repression against the separated area of chromatin and transcription machinery recruitment and establish a strict control system to ensure the expression of the *fbp1* gene only when three independent TFs are activated, which is probably a specific situation only under glucose starvation stress. This kind of Tup11/12 mediated antagonistic mechanisms are employed additionally for the TF binding events and further contribute to stress specificity and precise timing expression of the *fbp1* gene (described in Section 4 and Section 6).

The chromatin at the *fbp1* promoter region, fully opened for induction, was reconstituted to the repressive state, responding to the restoration of extracellular glucose concentration. When glucose was added back to the medium after glucose starvation stress, *fbp1* transcription was immediately shut down in 10 min, accompanied by dissociation of TFs, while the reconstitution of nucleosomes occurred slightly later [59] (Figure 2g,h). Generally, nucleosome assembly is facilitated by histone chaperones [60]. Screening of 10 potential histone chaperone genes identified that the histone chaperone Asf1 is involved in post-stress chromatin reconstitution to restore of the repressive state at the *fbp1* locus [59]. Interestingly, the *asf1* temperature sensitive mutant showed repression of *fbp1* transcription without chromatin reconstitution, suggesting that there are some repression controls, including a rapid reaction to stop unnecessary gene transcription and the subsequent restoration of chromatin to maintain chromatin architecture for the stable establishment of repressive state in non-stressed condition.

## 4. mlonRNAs-Transcription-Mediated Regulation of Chromatin and Transcription Factor Binding for *fbp1* Transcriptional Regulation

As described above, the cascade transcription of mlonRNAs upon glucose starvation induces the dynamic change of chromatin structure at *fbp1* upstream region (Figure 2c). Since chromatin remodeling events are generally associated with histone modifications, the role of mlonRNA transcription in its modulation was analyzed. Chromatin immunoprecipitation (ChIP) analysis with an antibody for histone modifications revealed that acetylation of histone H3 and H4 is induced by the progression of mlonRNAs-transcribing RNAPII [61]. These histones are modified by the histone acetyltransferase Gcn5 coupled with mlonRNA transcription. The bromodomain chromatin remodeler Snf22 recognizes this acetylation, converting the *fbp1* promoter chromatin to the open state [61,62] (Figure 3B). In addition to Snf22, the CHD1 family chromatin remodeler Hrp3 is also involved in this reaction. Interestingly, these chromatin remodelers double deletion mutant causes complete loss of chromatin remodeling at the *fbp1* region, suggesting that at least two pathways (Snd22 and Hrp3) are involved in this mlonRNA-transcription-mediated chromatin remodeling event [61]. 

Notably, mlonRNA-transcription-induced chromatin alteration was only observed in the *fbp1* upstream region, but not in the *fbp1* ORF, even though RNA polymerase transcribing mlonRNAs passes through the entire *fbp1* gene region and stops at the *fbp1* gene terminator (Figure 3A). This observation suggests that chromatin alteration mediated by mlonRNA transcription is restricted to a certain range from the mlonRNA TSS. Interestingly, considering the position of mlonRNA TSSs and regulatory elements (TF binding site, etc.), mlonRNAs always have downstream regulatory elements approximately 200 bp downstream from their TSSs (Figure 3A). The artificial modification of the distance between the mlonRNA TSS and downstream TF binding site by insertion or deletion in this genomic region revealed that the effective range of mlonRNA-transcription-mediated chromatin remodeling is restricted to within 290 bp. Thus, mlonRNAs work as short-range inducers for local chromatin alterations [63]. This is the reason why multiple mlonRNAs are transcribed in a stepwise manner and the cascade mlonRNAs transcription initiation with limited effective range provides the regulation to change only required local chromatin (around the TF binding site) and protects unnecessary chromatin alteration, which might cause undesirable transcription from inside the gene body [64]. 

Given the broadly employed cases in which local chromatin remodeling is induced by chromatin remodeling machineries recruited via TFs [65], it is unclear why the mlonRNA-mediated system for local and short-range chromatin regulation is adopted instead of the TF-mediated system. This suggests that other functional aspects of lncRNA transcription also play important roles in regulating chromatin transcription. Strikingly, mlonRNA investigations uncovered additional functions of lncRNA transcription, which are driven by transcription-generated DNA supercoils and transcribed lncRNA itself [62,66]. By integrating these multiple functions, lncRNAs can provide rigorous regulation through multiple aspects of genome-chromatin regulation.

Accompanying the reaction that separates DNA double strands (e.g., replication and transcription), DNA topology is altered by over- and under-winding of the DNA double helix, which affects the reactivity of DNA-related reactions [67]. Particularly during transcription, positively (over-winding) and negatively (under-winding) supercoiled DNA is generated ahead and behind the passing RNA polymerase, respectively [68]. Genome-wide DNA topological state analysis uncovered that negatively supercoiled DNA is accumulated at the gene promoter region, suggesting that the gene transcription generated upstream negative supercoils are still present and may involve regulation on the promoter region [69,70]. Some observations of in vitro assays, which show the influence of DNA topology in the nucleosome reconstitution and reactivity of chromatin remodeler suggested that DNA topology and chromatin regulation are closely related [71,72]. Given that lncRNAs are pervasively transcribed from the entire genome region, including regulatory elements in the intergenic region, it can be considered that lncRNA transcription induces and/or modulates DNA supercoils and is involved in the regulation of chromatin architecture. In *fbp1* regulation, a cascade of mlonRNA transcription is observed for regulating chromatin under both *fbp1* repressive and induced conditions (Figure 3A), and it could be hypothesized that negatively supercoiled DNA is always maintained at the *fbp1* promoter region, and it has important role for chromatin regulation. The effect of DNA supercoils on *fbp1* transcription regulation is analyzed by overexpression of topoisomerases which resolve DNA torsional stress or their direct recruitment at the local *fbp1* region to make the situation of over-resolution of DNA topology. It has been found that it causes aberrant positioning of nucleosomes at the *fbp1* promoter region, accompanied by the dysregulated *fbp1* transcription [66]. In wild-type cells, nucleosomes are uniformly distributed in the individual cells at specified locations in the *fbp1* upstream region under glucose-rich conditions. Under glucose starvation stress conditions, nucleosomes are largely removed by the chromatin remodeling event, but a few nucleosomes remain in the *fbp1* upstream region with repositioning from the original locations. These nucleosomes in topoisomerase overexpression/recruitment cells are destabilized and slightly shifted up or downstream and asynchronized in each individual cell [66] (Figure 3C). This irregulation of nucleosome positioning was also observed in the *prp3* gene promoter region, which is constitutively active without requirement of the glucose starvation signal [66]. These results indicate that DNA topology could be a critical determinant of nucleosome positioning, and that transcription-generated DNA supercoils at the promoter region by mlonRNAs or *prp3* gene transcription are required for maintaining proper nucleosome positioning. It was further suggested that the observed pervasive lncRNA transcriptions genome-wide, including mlonRNAs, might affect chromosome functions by affecting the local DNA topological state. 

The other functional aspects of lncRNA-mediated regulation are functions of transcribed lncRNA molecules. Many lncRNA investigations have revealed that transcribed lncRNA molecules directly interact with protein regulators and function as recruiters of specific functional protein molecules (e.g., chromatin modification machinery or transcription regulators) to a specific genome locus or decoy to sequester a certain protein for the regulation of many biological processes [18]. With regard to this, it has been found that transcribed mlonRNA molecules provide additional regulation involving Atf1 binding to the target locus [62]. Inhibition of mlonRNA production by a transcription inhibitor or deletion of the expected mlonRNA promoter region caused a reduction in Atf1 binding in the UAS1 region [62]. This defect in stable Atf1 binding by mlonRNA deletion was compensated by Tup11/12 deletion [62], indicating that this function is mediated by modulating Tup11/12 function. RNA immunoprecipitation analysis revealed that the transcribed mlonRNAs interacted with Tup11/12. Since Atf1 binding to UAS1 is required for inducing a cascade of mlonRNA transcription, transcribed mlonRNA-mediated regulation of Aft1 binding provides positive feedback regulation to achieve stable Atf1 binding for stable *fbp1* activation under glucose starvation stress (Figure 3D). Genome-wide Atf1 ChIP analysis found that this type of antagonistic regulation of Atf1 by Tup co-repressor and upstream transcribed lncRNA is also observed in some other loci and at least at *ght1* and *ght4* loci, the potential lncRNA molecules directly interact with Tup11/12, suggesting that this lncRNA mediated Aft1 regulation is globally adopted in fission yeast glucose starvation stress response.

## 5. mlonRNA Transcription Plays Roles in the Regulation of General Chromosomal Function in Fission Yeast Genome

With the increase in knowledge on the mechanism of mlonRNA function at the *fbp1* locus, it is important to assess whether mlonRNA is more broadly expressed in multiple gene loci and has a pivotal role in the regulation of chromosome function via modulation of chromatin structure other than in the transcription process. As shown above, some genome-wide analyses (RNA-seq and ChIP-seq to explore the genome loci with lncRNA transcription as well as alteration of chromatin structure/TF binding) have provided evidence of the existence of other mlonRNA-type lncRNAs in the regulation of glucose starvation stress [58,62]. In addition, the identification of the sequence of the essential transcription initiation element for mlonRNA-c enabled further investigation for exploring mlonRNA-type lncRNAs in different loci and, if any, in a variety of chromosome regulation processes. Comprehensive segmentation and replacement of the short DNA sequence from the mlonRNA–c TSS toward its upstream region identified a DNA segment specifically required for mlonRNA–c transcription induction located ~100 bp upstream from its TSS [63]. The single-base one-by-one mutation of this short DNA segment identified a 9-nucleotide sequence driving mlonRNA–c transcription (5′- A/T A/C T T/G A T/C/G G T A/G-3′), which is termed as *mlon-box* [73]. Mapping of the *mlon-box* sequence in the fission yeast genome showed enrichment of this sequence near upstream from the annotated gene TSS, indicating that this sequence works for the transcription initiation of the other loci. More interestingly, a correlation that showed close localization with the meiotic recombination hotspot has also been found. In meiosis, homologous recombination is an essential process that is initiated by the induction of DNA double-strand breaks (DSBs) at certain genome locations [74]. The selection of DSB sites is not completely random, and open chromatin regions, such as nucleosome-free regions at the promoter, are frequently selected because they are more accessible for the DSB-catalyzing enzyme [75]. As such, the correlation of recombination “hotspot” and *mlon-box* sequence arose interesting hypothesis that mlonRNAs responding in meiosis works for determination of DSB site via the modulation of chromatin architecture. 

The hypothesis that mlonRNA-transcription-mediated chromatin regulation in meiosis contributes to regulating DSB site selection was first assessed by inserting the *mlon-box* sequence into the well-characterized meiotic recombination hotspot *ade6-M26* [76] to test whether mlonRNA accelerates chromatin remodeling followed by meiotic recombination. The *M26* mutation is a nonsense mutation in *ade6*, creating a CRE-like heptanucleotide sequence 5′-ATGACGT-3′, to which the transcription factor Atf1 binds and activates meiotic recombination [42]. Atf1 binding induces *M26* transcription from the *M26* mutation site in the *ade6* ORF and chromatin remodeling around the *M26* site for the induction of meiotic recombination, which is similar to *fbp1* regulation [52,77]. To mimic the situation in the *fbp1* regulation, in which the *mlon-box* is located 200 bp downstream from UAS1 comprising an Atf1 binding sequence in *fbp1*, the *mlon-box* was placed 200 bp downstream from the *M26* mutation site by replacing the same size of the *ade6* ORF (Figure 4A). The insertion caused additional activation of transcription from the insertion site and, more importantly, stimulated meiotic recombination via local chromatin remodeling [73] (Figure 4A). Then, the naturally encoded *mlon-box* sites, which locate close to the recombination hotspot in the *S. pombe* genome, are focused. One such genomic location, the *SPBC24C6.09c* upstream region, has strong recombination hotspot activity and strong meiosis-induced transcription [73]. Mutation of the natural *mlon-box* sequence at the *SPBC24C6.09c* site dramatically reduces transcription and DSB induction for recombination. Interestingly, this impaired DSB formation was accompanied by a defect in *mlon-box*-dependent chromatin opening at the recombination site (Figure 4B). These results suggest a universal role of mlonRNA transcription, and that mlonRNA-transcrirption-mediated chromatin remodeling has the potential to regulate many aspects of chromosome functions, including transcription and recombination. 

## 6. Tup Co-Repressor Mediated Multi-Layered Regulations

As summarized above, the complex coordination of TFs and mlonRNAs in the regulation of chromatin structure establishes the strict regulation of *fbp1* gene transcriptional induction. In this process, the most interesting phenomenon is that *fbp1* transcription is ensured to be limited only to glucose starvation stress as the transcription co-repressor Tup11 and Tup12, a double deletion mutant, induces *fbp1* transcription under non-glucose-starvation stress conditions [52]. ChIP experiments to analyze Tup11/12 binding to the *fbp1* locus showed that Tup11/12 localize to the *fbp1* promoter region with peaks at UAS1 and UAS2, and the binding intensity was higher in the *fbp1* activated condition, despite the transcription repressor [36]. These suggest that the Tup co-repressor is not just a “repressor” but is also a “regulator” of the transcription to fine-tune the timing of the target gene transcription induction in the qualified condition. In the case of *fbp1* regulation, Tup11/12 repressed Atf1 binding, chromatin remodeling, and transcription machinery recruitment into the *fbp1* promoter (Section 3 and Section 4) (Figure 5A and Figure 3D). These repressions are antagonized by mlonRNAs, and the independently recruited three TFs, enabling them to proceed in a step-by-step manner for *fbp1* induction. Recently, additional Tup-mediated repressions have been identified, and it has been shown that genome architectures, including the proximity of TF binding sites and three-dimensional chromatin loops, are precisely placed or modulated to antagonize Tup-mediated repressions. Integrating these multi-layered regulations with multiple Tup-repressions probably contributes to strict control of *fbp1* transcription in a stress-specific manner.

### 6.1. Local Proximity of Two TF-Binding Motifs Integrates Distinct Signal Pathway on Genome for Antagonizing Tup-Mediated Inhibition of Atf1 and Rst2 Binding

Around the UAS1 region, an additional Rst2 binding CT-rich sequence was identified, and thus, the two binding sites for the distinct TFs, Atf1, and Rst2, were placed in close proximity (45 bp apart) (Figure 5B). Indirect end-labeling analysis of MNase-digested chromatin DNA revealed that a couple of MNase-sensitive bands appeared in the UAS1 region, and these nuclease-sensitive bands were dependent on the activation of both Atf1 and Rst2 [78]. Moreover, by placing the two TF binding sites close to each other, the binding of Atf1 and Rst2 in the UAS1 region was reciprocally stabilized [78] (Figure 5B(a)). Atf1 and Rst2 are controlled by their phosphorylation states under distinct signaling pathways, the MAPK and PKA [32,39,40], and these results suggest a previously unappreciated mechanism by which two TF binding sites in close proximity integrate two independent signaling pathways, thereby behaving as a hub for signal integration [78]. Importantly, the destabilization of independent Atf1 and Rst2 binding is mediated by the function of Tup11/12 co-repressors [78] (Figure 5B(b,c)). This indicates that Tup co-repressors generate a type of stress-type sensing, where only the stress in which both MAPK and PKA pathways are activated, allowing for the next step in the regulation of *fbp1* transcription. Sequential-ChIP analysis of Atf1 and Rst2 followed by next-generation sequencing analysis to find genome-wide co-binding sites identified 536 and 837 peak signals in glucose-rich and starvation conditions, respectively [78], indicating that several genes are regulated by the integration of two signaling pathways. As a highly enriched and highly augmented (in glucose starvation) region, the *ght4* hexose transporter gene was analyzed, and it was revealed that the *ght4* gene promoter carries two sequences arranged in tandem containing both putative CRE (Atf1-binding site) and CT-rich sequences (Rst2 binding sequence) located 33 and 42 bp apart, respectively. In this region, Atf1 and Rst2 were interdependently bound, as in the *fbp1* UAS1 region.

### 6.2. Role of Genome-Local Loop Structure in the Precise TF-Binding and Transcriptional Initiation

In the last couple of decades, the development of chromosome conformation capture (3C) technology and its derivatives have allowed the mapping of the three-dimensional genome elements, and it is now commonly accepted that modulation of chromatin loop structure is an additional layer of transcription regulation [79,80,81,82]. In addition, in *fbp1* regulation, it has been found that a local chromatin loop is generated and has a critical function. With the unique Tup11/12 distribution of their binding at the *fbp1* region, the other three TFs (Atf1, CBF, and Rst2) also exhibit two peaks at UAS1 and UAS2 [43]. These unique binding distributions of all TFs and co-repressors suggest that UAS1 and UAS2 are placed in close proximity three-dimensionally during the *fbp1* gene activation processes. This was indeed the case, since the 3C technology detected a significant interaction between UAS1 and UAS2, and this chromatin loop structure is termed the UAS-loop [43]. This UAS loop forms after chromatin opening in the UAS1-UAS2 region and this formation requires all three TFs. Further analysis revealed that the UAS loop plays an important role in regulating Rst2 binding to UAS2. During *fbp1* activation, Rst2 is first recruited to the CT-rich region near UAS1, which is stabilized by the coordinated proximal binding of Atf1 to UAS1, and the Rst2 is subsequently delivered to UAS2 through the UAS loop structure that brings UAS1 and UAS2 into close proximity three-dimensionally (Figure 5C). Interestingly, Tup11/12 co-repressors suppressed the direct binding of Rst2 to UAS2, however, this suppression was counteracted by the delivery of Rst2 bound to UAS1 (Figure 5C). Note that Rst2 binding to UAS2 is required for the final step of *fbp1* activation, which recruits the transcription machinery to the *fbp1* promoter. Because UAS loop formation requires three TF bindings at the *fbp1* locus, this counteractive regulation between Tup co-repressor mediated repression and UAS loop mediated delivery plays a role to decide the proper timing of *fbp1* transcription activation (i.e., *fbp1* is expressed only when all three TFs are activated by the signaling pathway) and might contribute to providing stress specificity of the *fbp1* transcription activation.

## 7. Summary and Perspective

In this review, we summarize the transcription regulation mechanism organized by the multiple layered regulations on the local genome architectures, focusing on the fission yeast *fbp1* gene as a model system. The layers of hierarchical genome architectures include three-dimensional genome structures, chromatin structure, DNA topology, the position of TFs binding sequences, and the coordination of these regulations establishes strict regulation of *fbp1* transcription induction. In particular, Tup co-repressors have multiple repression functions, and multi-layered repression and antagonization mechanisms provide transcription stringency to achieve tight and condition specific control. Gene expression regulation should be diverse depending on the encoding protein function. Therefore, the gene that has the same transcription regulation system as the *fbp1* gene is not known. However, some genome-wide analyses have shown evidence that some of the *fbp1* gene regulatory layers are commonly used in many other genes, and it is expected that different combinations of the transcription regulation layers provide divergence for the gene-specific regulation of transcription. Although intensive analyses have been conducted using this model system, some unsolved important questions remain. First, although the roles of Tup11/12 as a modulator of transcription, but not simple repressor, have been established [36,43,52,78], the molecular mechanisms by which Tup11/12 co-repressors establish multiple different repressions and their antagonizations have not been fully elucidated. Second, while the consensus sequence of the *mlon-box* was clarified and the general roles of mlonRNA transcription in the regulation of chromosome functions (not only for transcription but also for meiotic recombination) have been established [73], the protein complex of mlonRNA transcribing RNA polymerase II (RNAPII) required for the mlonRNA-specific function (i.e., induction of chromatin remodeling not observed in regular mRNA transcription) has not been identified. An intriguing possibility is that RNAPII initiating mlonRNA transcription carries unique accessory subunit(s) that possess histone acetyltransferase activity to induce local histone acetylation around TSSs and dissociate from the initiation complex after promoter clearance [83]. Such histone modifications may recruit ATP-dependent chromatin remodelers [9]. This hypothesis is supported by the observation that histone acetylation is gradually induced upstream of *fbp1* during glucose starvation, and a histone acetyltransferase, Gcn5, and an ATP-dependent chromatin remodeler, Snf22, are required for chromatin remodeling upstream of *fbp1* [61,62]. If such a novel RNAPII complex exists, it will be important to understand the mechanism of selective targeting to the sites of mlonRNA transcription initiation, but not regular mRNA genes TSS. The third question is whether the chromatin modulation phenomenon accompanied by mlonRNA synthesis is conserved in higher animal cells. This may be addressed by the analysis of the identified protein complex for mlonRNA initiation because it is expected that the *mlon-box* sequence might not be conserved in other organisms, but proteins involved in mlonRNA initiation might be conserved in other organisms.

## Figures and Tables

**Figure 1 biomolecules-12-01642-f001:**
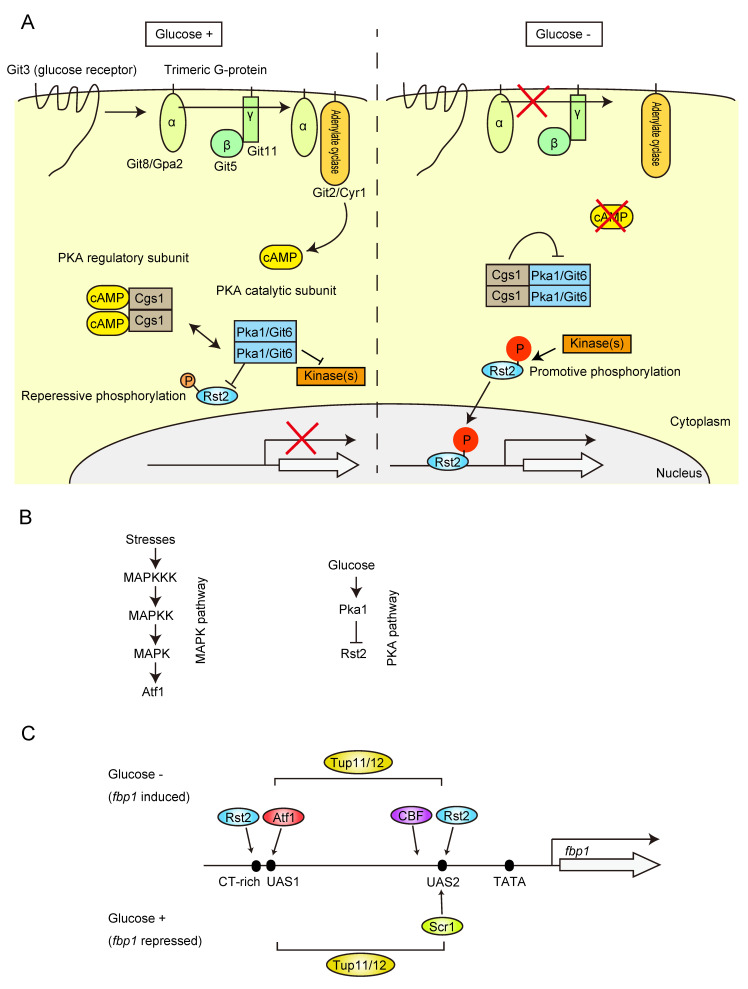
Signal pathways and TFs involved in the regulation of *fbp1* gene. (**A**) Schematic representation of glucose sensing and signal pathway. Presence of environmental glucose is sensed by the seven-transmembrane receptor Git3 coupled with trimeric G protein, Gpa2-Git5-Git11. Activated G protein further activates adenylate cyclase, Cyr1, which in turn catalyzes cAMP synthesis. Thus, the presence of environmental glucose is signaled as an inter-cellular cAMP concentration. The resultant cAMP binds to the regulatory subunit of PKA (Cgs1) and cAMP-bound Cgs1 is released from catalytic subunit of PKA (Pka1) to activate Pka1. Phosphorylation of TF Rst2 by Pka1 (Repressive phosphorylation) inhibits hyperphosphorylation of Rst2 by other kinase(s) and following transition of Rst2 to the nucleus. Loss of cAMP production by depleting glucose represses Pka1 activity by forming complex with the inhibitory subunit, Cgs1, resulting loss of repressive phosphorylation of Rst2 which in turn results in hyperphosphorylation of Rst2, and Rst2 is imported to the nucleus for transcription activation. (**B**) Glucose starvation stress activates MAPK pathway. This results in the phosphorylation of TF Atf1 and induces transcription activation of its target genes. (**C**) Schematic representation of *fbp1* upstream regulatory sequence and the targeting TFs. The upstream activation sites 1 and 2 (UAS1 and UAS2) are binding sites for Atf1 and Rst2, respectively. The two TFs, Atf1 and Rst2, are regulated under MAPK and PKA pathways, respectively. Rst2 also targets the CT-rich sequence near the UAS1. CBF (CCAAT-binding factor) targets the sequence near the UAS2. UAS2 is also a target site of Scr1 in glucose rich sequence. UAS2 is reciprocally occupied by Scr1 and Rst2 for the repression and activation of *fbp1* gene depending on the environmental condition. Tup co-repressor locates at *fbp1* upstream region across UAS1 to UAS2 in both repressive and depressive conditions to provide glucose specific regulation of *fbp1* transcription induction.

**Figure 2 biomolecules-12-01642-f002:**
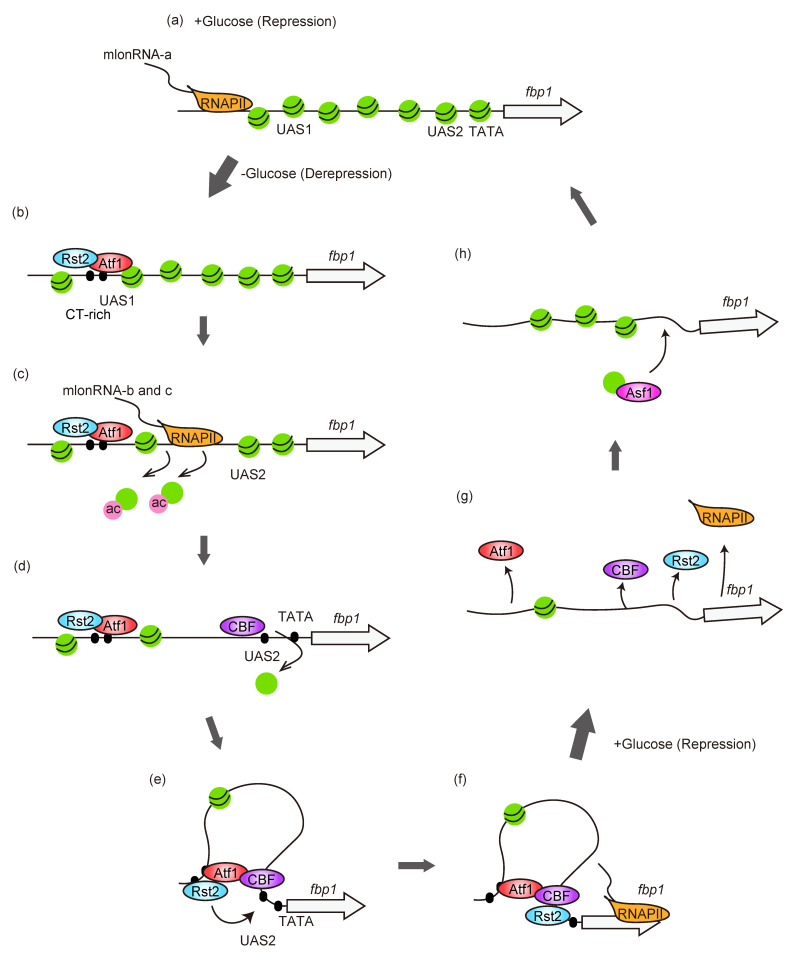
Dynamics of genome-chromatin architectures in *fbp1* transcription regulation. (**a**) Repressive chromatin is assembled at the *fbp1* promoter region in glucose rich condition. The longest mlonRNA–a is weakly transcribed in this condition. (**b**) In response to glucose depletion stress, Atf1 and Rst2 are activated through signaling pathways and bind to the upstream *fbp1* regulatory region, UAS1 and CT-rich motif, respectively. The binding of these factors induces remodeling of chromatin around UAS1 to be the open state. (**c**) Atf1 binding induces stepwise transcriptions of mlonRNAs–b and –c, which induces histone acetylation followed by chromatin remodeling around UAS1-UAS2 region. (**d**) When the chromatin around UAS2 becomes accessible, CBF binds to near the UAS2 and induces chromatin opening around *fbp1* TATA box. (**e**) When three TFs are all bound to the *fbp1* region, local chromatin loop structure is formed presumably by the interaction of these TFs. This allows the targeting of Rst2 to the functional binding site, UAS2, by delivering from CT-rich motif near the UAS1. (**f**) UAS2 bound Rst2 facilitates recruitment of transcription machinery to the *fbp1* TATA box and induces massive transcription of *fbp1* mRNA. (**g**) After releasing from the stress, the *fbp1* transcription is immediately stopped simultaneously with the dissociation of TFs. (**h**) The nucleosomes are then reconstituted by histone chaperon Asf1, which returns to the repressive chromatin.

**Figure 3 biomolecules-12-01642-f003:**
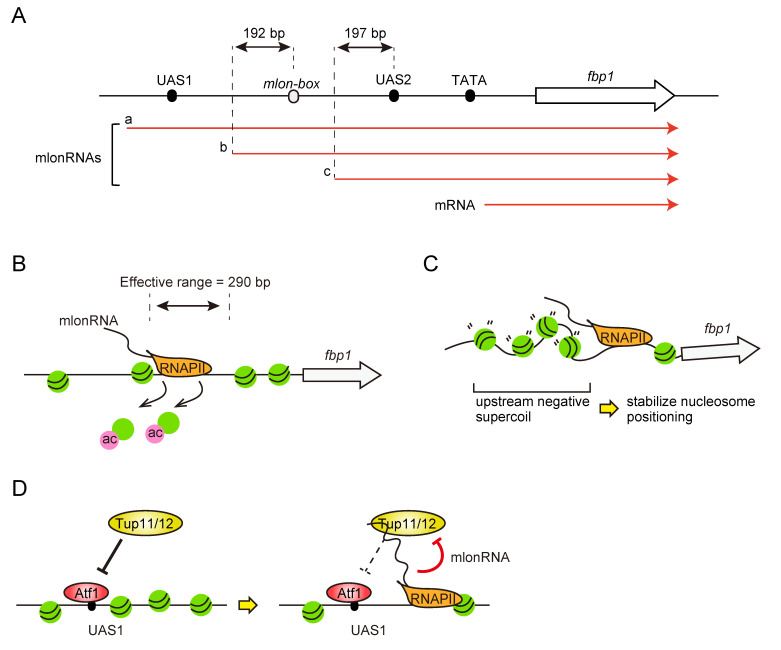
mlonRNA-transcription-mediated regulations of chromatin structure and transcription factor binding. (**A**) Schematic representation of lncRNAs transcribed from *fbp1* locus. In glucose rich condition, the longest mlonRNA (mlonRNA–a) are weakly expressed. At the early time of glucose starvation (10–20 min of glucose starvation), mlonRNA–b and –c are progressively expressed. At 60–180 min of glucose starvation, *fbp1*–mRNA is massively induced. mlonRNA–c initiation sequence, *mlon-box*, is located around 100 bp upstream from mlonRNA–c TSS. Note that both mlonRNA–b and –c have DNA element (*mlon-box* and UAS2, respectively) around 200 bp downstream from their TSS. (**B**) mlonRNA-transcription-mediated chromatin remodeling. mlonRNA transcription induces histone acetylation by histone acetyl transferase Gcn5. The acetylated nucleosomes are removed or repositioned by chromatin remodeler(s). The effective range of mlonRNA transcription induced chromatin remodeling is restricted within 290 bp. This allows limited chromatin opening for just downstream DNA elements, in this case, *mlon-box* and UAS2 for mlonRNA–b and –c, respectively. (**C**) mlonRNA transcription generated DNA supercoil stabilizes nucleosome positioning. In both *fbp1* repressed and derepressed conditions, transcription of either mlonRNAs or *fbp1* mRNA always occurs, and this probably induces negative supercoil at *fbp1* promoter region. Because excess resolution of DNA supercoil by topoisomerase causes asynchronous irregular nucleosome positioning in each individual cell, mlonRNA (or *fbp1* mRNA) generated DNA supercoil might be required for stable maintenance of positioned nucleosomes at proper locations. (**D**) The function of transcribed mlonRNA molecule. Transcribed mlonRNAs interact with Tup11/12 co-repressors and this antagonizes the repressive function of Tup11/12, which inhibits stable association of Atf1 to UAS1.

**Figure 4 biomolecules-12-01642-f004:**
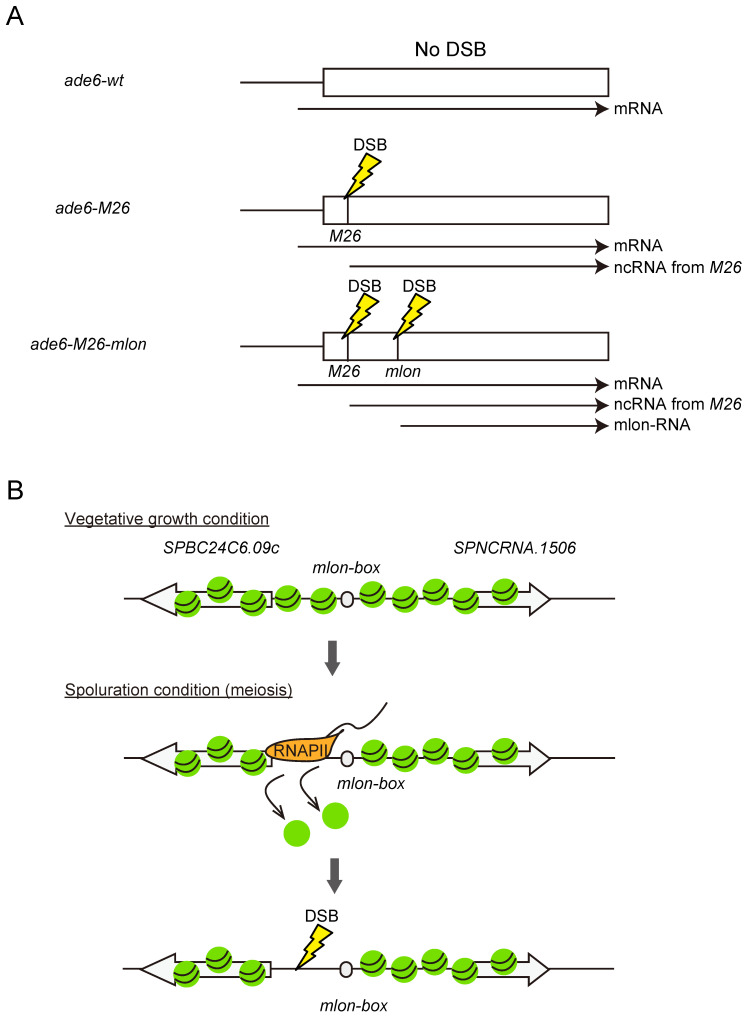
Mechanism of mlonRNA-transcription-induced meiotic recombination hotspot. (**A**) Schematic representation of *ade6* gene locus. Position of *M26* mutation and the insertion point of the *cis*-element for the shortest mlonRNA (mlonRNA–c) are indicated by vertical lines. The *ade6* mRNA and transcripts induced from *M26* site or *cis*-element insertion site are indicated by arrows. The DSB sites that initiate meiotic recombination are shown. Additional mlonRNA initiation sequence induces mlonRNA transcription coupled with chromatin remodeling, which facilitates the DSB induction. (**B**) Natural recombination hotspot induced by *mlon-box* driven transcription in *S. pombe* genome. *SPBC24C6.09c* gene locus has *mlon-box* sequence. Transcription from this *mlon-box* is not active in vegetative cell growth, but it is activated in response to the meiosis induction. This mlonRNA-type transcription generates open chromatin and in turn induces DSB.

**Figure 5 biomolecules-12-01642-f005:**
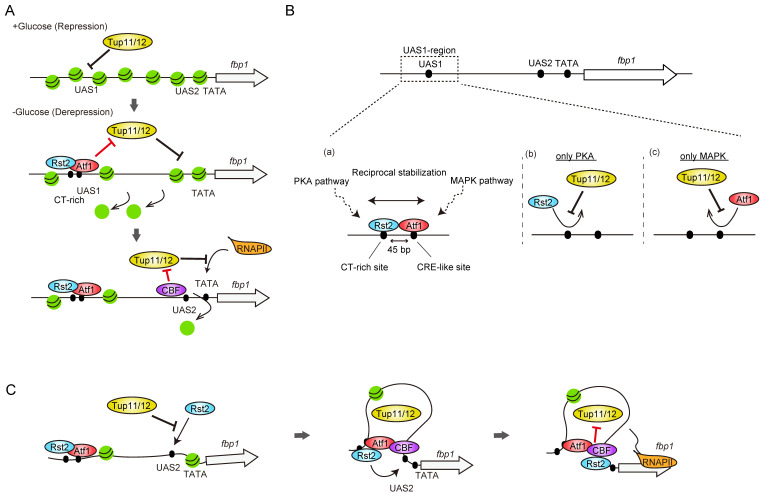
Tup11/12 mediated multi-layered repression and antagonization system for stress specific induction of *fbp1* transcription. (**A**) Tup11/12 mediated regulations on the chromatin configurations. In glucose rich conditions, Tup11/12 repress the chromatin remodeling at UAS1-UAS2 region. Repressive chromatin at this region is counteracted by the Atf1 in the initial time point after glucose starvation. Tup still represses chromatin remodeling at TATA box region. By the chromatin opening around UAS2, CBF can be recruited, and it antagonizes second Tup11/12 chromatin repression, induction of chromatin opening at TATA box. Then, Tup11/12 next represses recruitment of transcription machinery at *fbp1* TATA box. This is finally counteracted by UAS2 bound Rst2 (continue to panel (**C**)). (**B**) Reciprocal stabilization of Atf1 and Rst2 at UAS1 region. The binding of Atf1 and Rst2 at UAS1 and CT-rich motif is also regulated by Tup11/12 mediated repression. At this region, the two TF binding DNA elements are located in close proximity (45 bp apart). The independent binding of Aft1 or Rst2 alone is inhibited by Tup11/12 (b,c). When both Atf1 and Rst2 are activated by two different signaling pathways (PKA and MAPK), these two factors can be stably bound at this region, probably due to the stabilization by mutual interaction with each other (a). (**C**) Local chromatin loop (UAS loop) and Tup mediated repression against Rst2 binding. By the reciprocal stabilization mechanism of Rst2 around UAS1 region with Atf1, it can bind to the CT-rich motif, but the binding of another essential binding site, UAS2, is inhibited by Tup11/12. This is counteracted by the delivery of Rst2 from CT-rich motif near the UAS1 to UAS2 via the local chromatin loop structure, UAS loop. After chromatin opening around UAS2 followed by CBF recruitment, UAS loop is formed, and this allows stable association of Rst2 to UAS2 by antagonizing Tup11/12 mediated repression. UAS2 bound Rst2 antagonizes another Tup11/12 repression of the recruitment of transcription machinery to TATA box, resulting massive induction of *fbp1* mRNA transcription.

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
