# Peer review of "Multi-Layered Regulations on the Chromatin Architectures: Establishing the Tight and Specific Responses of Fission Yeast fbp1 Gene Transcription"

_biomolecules, 2022, doi:10.3390/biom12111642_

Round 1

Reviewer 1 Report

In the manuscript Ryuta Asada and Kouji Hirota review the complex regulatory system controlling transcription of fbp1 gene in well regarded model organism fission yeast Schizosaccharomyces pombe with particular focus on integration of various levels of regulation and their effects on chromatin organization of fbp1 promoter. The fbp1 gene encodes fructose-1,6-biphosphatase, an enzyme catalyzing a rate limiting step of gluconeogenesis. This gene has been extensively researched as model for understanding stringent transcriptional regulation - fbp1 gene being under tight glucose repression. The stringent control of fbp1activity is achieved by a complex regulatory cascade initiated with signal transduction pathways responsive to glucose stress and controlling activity of several sequence specific transcription factors. Atf1 ATF/CREB protein, Zn-finger proteins Rst2 and Scr1, and CCAAT binding factor CBF localize to fbp1 promoter and regulate its activity. The localization of TFs to fbp1 promoter is accompanied by sequential remodeling of promoter chromatin by Pol2 traversing the promoter region during transcription of lncRNAs that converts chromatin into open configuration and induces fbp1 expression. The stringent control of fbp1 activation is ensured by Tup repressors that oppose chromatin remodeling of fbp1 promoter and prevent its spurious activation 

The review provides comprehensive summary of the fbp1 gene multilayered control and analyzes recent advances in our understanding of chromatin architecture in fbp1 regulation. The manuscript is well-illustrated, properly organized and referenced, is generally suitable for Biomolecules readership. However, the text clarity is inconsistent, several sections, particularly figure legends require additional editing.

Specific suggestions to authors:

Please consider following revisions

3D Three-dimensional                                                                                             lines 42, 43, 62

Seventh transmembrane receptor                                                                             lines 95, 151

reseptor (receptor) misspelling                                                                                  Figure 1a top

G protein further activates adenylate                                                                         line 152

represses of Pka1                                                                                                        line 157

and Rst2 imports                                                                                                         line 159

Please consider rewording for clarity - “Tup co-repressor locates at fbp1 upstream region across UAS1 to UAS2 in both repressive and derepressive conditions for critical regulation to provide glucose specific regulation of fbp1 transcription induction.”                                       line 168           

The important roles of chromatin consisting of nucleosome arrays in transcriptional regulation have already been highlighted.                                                                          line 172

The chromatin alteration remodeling kinetics were                                                  line 182

Please consider rewording “from the 5’ region to the 3’ region of fbp1 upstream “

perhaps with  “from the 5’ to the 3’ end of fbp1 upstream region”                          line 187

it needs Rst2 binding to UAS2    ->   Rst2 binding to UAS2 is required                       line 215

Asf1 is involved in post-stress chromatin reconstitution to restore of the repressive state at the fbp1locus                                                                                                                      line 234

Lines 235-239 unclear, please consider rewording,

mlonRNA-a is slightly weakly transcribed in this condition.                                       line 243

and Rst2 are activated by the response of signaling pathway                                    line 244

of chromatin around UAS1 to be the open state.                                                       line 246

CBF targets to the binding binds site near the UAS2                                                  line 249

transcription is immediately stopped accompanying simultaneously with                line 254

Asf1, which is returnes to the repressive chromatin.                                                 line 256

majority of transcripts are do not originate from protein-coding                              line 260

please cite relevant paper                                                                                         line 315

nucleosomes are usually well synchronized (regularly spaced?) at specified loca-    

line 345

are destabilized and asynchronized irregularly spaced with slight                             line 350

transcription of either mlonRNAs or fbp1 mRNA is always occurs happened            line 394

by topoisomerase causes asynchronous irregular nucleosome positioning,             line 396

this antagonizes against the repressive function of Tup11/12,                                  line 400

The hypothesis that for mlonRNA-mediated chromatin regulation in meiosis for regulates DSB site selection was first assessed                                                                          line 431

Unclear meaning: “Then, it is focused on the natural mlon-box sites flanking the recombination hotspot. “                                                                                                                       line 444

of fbp1 gene transcription induction                                                                          line 467

is that fbp1 transcription is ensured to be expressed limited only to glucose starvation      

line 469

showed that Tup11/12 distributes localize to the fbp1 promoter region                  line 472

Local proximity of two TF-binding sequences motifs integrates distinct signal pathways on genome for antagonizing Tup-mediated inhibition of Atf1 and Rst2 binding                     line 486

Line 533-535;  Please consider rewording for better clarity.

Then, Tup11/12 still express the next represses recruitment of transcription machinery at fbp1 TATA box                                                                                                            line 546

to the stabilization by mutual interaction each other                                                line 553

another Tup11/12 repression against of the recruitment of…                                  line 560

Author Response

In the manuscript Ryuta Asada and Kouji Hirota review the complex regulatory system controlling transcription of fbp1 gene in well regarded model organism fission yeast Schizosaccharomyces pombe with particular focus on integration of various levels of regulation and their effects on chromatin organization of fbp1 promoter. The fbp1 gene encodes fructose-1,6-biphosphatase, an enzyme catalyzing a rate limiting step of gluconeogenesis. This gene has been extensively researched as model for understanding stringent transcriptional regulation - fbp1 gene being under tight glucose repression. The stringent control of fbp1activity is achieved by a complex regulatory cascade initiated with signal transduction pathways responsive to glucose stress and controlling activity of several sequence specific transcription factors. Atf1 ATF/CREB protein, Zn-finger proteins Rst2 and Scr1, and CCAAT binding factor CBF localize to fbp1 promoter and regulate its activity. The localization of TFs to fbp1 promoter is accompanied by sequential remodeling of promoter chromatin by Pol2 traversing the promoter region during transcription of lncRNAs that converts chromatin into open configuration and induces fbp1 expression. The stringent control of fbp1 activation is ensured by Tup repressors that oppose chromatin remodeling of fbp1 promoter and prevent its spurious activation 

The review provides comprehensive summary of the fbp1 gene multilayered control and analyzes recent advances in our understanding of chromatin architecture in fbp1 regulation. The manuscript is well-illustrated, properly organized and referenced, is generally suitable for Biomolecules readership. However, the text clarity is inconsistent, several sections, particularly figure legends require additional editing.

 (Response) Thank you for your overall positive judgment for your manuscript. We revised all issues raised bellow.

Specific suggestions to authors:

Please consider following revisions

3D Three-dimensional                                                                                             lines 42, 43, 62

(Response) Thank you. We fixed these errors. All terms ‘3D’ was replaced by ‘three-dimensional’ in text.

Seventh transmembrane receptor                                                                             lines 95, 151 (Response) Thanks. We fixed these errors. All terms ‘Seventh transmembrane receptor’ was replaced by ‘Seven-transmembrane receptor         ’ in text.

reseptor (receptor) misspelling                                                                                  Figure 1a top

(Response) Thanks. We fixed this error.

G protein further activates adenylate                                                                         line 152

(Response) Thanks. We fixed this error. The terms ‘activate’ was replaced by ‘activates’ in text.

represses of Pka1                                                                                                        line 157

(Response) Thanks. We fixed this error. the term ‘of’ was removed.

and Rst2 imports                                                                                                         line 159

(Response) Thanks. We fixed this error. the sentence ‘Rst2 imports’ was changed to ‘Rst2 is imported’

Please consider rewording for clarity - “Tup co-repressor locates at fbp1 upstream region across UAS1 to UAS2 in both repressive and derepressive conditions for critical regulation to provide glucose specific regulation of fbp1 transcription induction.”                                       line 168           

 (Response) Thanks. We fixed this sentence. We reworded as follow. “Tup co-repressor locates at fbp1 upstream region across UAS1 to UAS2 in both repressive and derepressive conditions to provide glucose specific regulation of fbp1 transcription induction.”

The important roles of chromatin consisting of nucleosome arrays in transcriptional regulation have already been highlighted.                                                                          line 172

 (Response) Thanks. We reworded this sentence as follows. “The importance of chromatin structure in transcriptional regulation had already been highlighted. ”

The chromatin alteration remodeling kinetics were                                                  line 182

 (Response) Thanks. We reworded this sentence as follows. “The chromatin remodeling kinetics were ”

Please consider rewording “from the 5’ region to the 3’ region of fbp1 upstream “

perhaps with  “from the 5’ to the 3’ end of fbp1 upstream region”                          line 187

  (Response) Thanks. This sentence is removed following comments from referee 2.

it needs Rst2 binding to UAS2    ->   Rst2 binding to UAS2 is required                       line 215

   (Response) Thanks. We reworded this sentence as suggested.

Asf1 is involved in post-stress chromatin reconstitution to restore of the repressive state at the fbp1locus                                                                                                                      line 234

   (Response) Thanks. We reworded this sentence as suggested.

Lines 235-239 unclear, please consider rewording,

(Response) Thanks. We reworded this sentence as follows. “Interestingly, the asf1 temperature sensitive mutant showed repression of fbp1 tran-scription without chromatin reconstitution, suggesting that there are some repression controls, including a rapid reaction to stop unnecessary gene transcription and the sub-sequent restoration of chromatin to maintain chromatin architecture for the stable establishment of repressive state in non-stressed condition.”

mlonRNA-a is slightly weakly transcribed in this condition.                                       line 243

 (Response) Thanks. We reworded this sentence as follows. “mlonRNA-a is weakly transcribed in this condition.”

and Rst2 are activated by the response of signaling pathway                                    line 244

 (Response) Thanks. We reworded this sentence as follows. “and Rst2 are activated through signaling pathways”

of chromatin around UAS1 to be the open state.                                                       line 246

   (Response) Thanks. We reworded this sentence as suggested.

CBF targets to the binding binds site near the UAS2                                                  line 249

    (Response) Thanks. We reworded this sentence as suggested.

transcription is immediately stopped accompanying simultaneously with                line 254

     (Response) Thanks. We reworded this sentence as suggested.

Asf1, which is returnes to the repressive chromatin.                                                 line 256

  (Response) Thanks. We reworded this sentence as suggested.

majority of transcripts are do not originate from protein-coding                              line 260

      (Response) Thanks. We reworded this sentence as suggested. Also, following the comment from referee 2, we moved this sentence to the introduction section.

please cite relevant paper                                                                                         line 315

      (Response) Thanks. We cited appropriate papers here.

nucleosomes are usually well synchronized (regularly spaced?) at specified loca-    

line 345

are destabilized and asynchronized irregularly spaced with slight                             line 350

 (Response) In these two sentences, we wanted to say that individual cells have the uniform chromatin configuration in wild type cells and Topoisomerase-overexpressed cells show destabilized and asynchronized chromatin structure in the each individual cells.

According to this suggestion, we rephrased as follows.

“In wild-type cells, nucleosomes are uniformly distributed in the individual cells at specified locations in the fbp1 upstream region under glucose-rich conditions.”

“are destabilized and slightly shifted up or downstream and asynchronized in each individual cell”

transcription of either mlonRNAs or fbp1 mRNA is always occurs happened            line 394

   (Response) Thanks. We reworded this sentence as suggested.

by topoisomerase causes asynchronous irregular nucleosome positioning,             line 396

   (Response) Thanks. We rewarded this sentence as suggested.

this antagonizes against the repressive function of Tup11/12,                                  line 400

   (Response) Thanks. We reworded this sentence as suggested.

The hypothesis that for mlonRNA-mediated chromatin regulation in meiosis for regulates DSB site selection was first assessed                                                                          line 431

 (Response) Thanks. We reworded this sentence as follows.

“The hypothesis that mlonRNA-mediated chromatin regulation in meiosis contributes to regulating DSB site selection”

Unclear meaning: “Then, it is focused on the natural mlon-box sites flanking the recombination hotspot. “  

 (Response) Thanks. We reworded this sentence as follows.

“Then, the naturally encoded mlon-box sites, which locate close to the recombination hotspot in the S. pombe genome, are focused.”

                                                                                                                     line 444

of fbp1 gene transcription induction                                                                          line 467

 (Response) Thanks. We reworded this sentence as follows.

 “of fbp1 gene transcriptional induction.”

is that fbp1 transcription is ensured to be expressed limited only to glucose starvation      

line 469

 (Response) Thanks. We reworded this sentence as suggested above.

showed that Tup11/12 distributes localize to the fbp1 promoter region                  line 472

 (Response) Thanks. We reworded this sentence as suggested above.

Local proximity of two TF-binding sequences motifs integrates distinct signal pathways on genome for antagonizing Tup-mediated inhibition of Atf1 and Rst2 binding                     line 486

 (Response) Thanks. We reworded this sentence as suggested above.

Line 533-535;  Please consider rewording for better clarity.

 (Response) Thanks. We reworded this sentence as follows.

“Because UAS loop formation requires three TF binding at the fbp1 locus, this counteractive regulation between Tup co-repressor mediated repression and UAS loop mediated delivery plays a role to decide the proper timing of fbp1 transcription activation (i.e., fbp1 is expressed only when all three TFs are activated by the signaling pathway) and might contribute to providing stress specificity of the fbp1 transcription activation.”

Then, Tup11/12 still express the next represses recruitment of transcription machinery at fbp1 TATA box        line 546

(Response) Thanks. We reworded this sentence as suggested above.                                                                                                  

to the stabilization by mutual interaction each other                                                line 553

(Response) Thanks. We reworded this sentence as suggested above.

another Tup11/12 repression against of the recruitment of…                                  line 560

(Response) Thanks. We reworded this sentence as suggested above.

Reviewer 2 Report

Comments to the manuscript written by Asada and Hirota:

In this review manuscript, the authors review the regulatory mechanisms of fbp1 gene expression in fission yeast, encompassing many years of research, including their own work.

Early work by Dr. Hoffman and by Dr. Yamamoto and by the authors’ subsequent works have revealed that the fbp1 gene expression is regulated by multilayered processes, which involve multiple actions of transcriptional activators and repressors and the arrangement of transcriptional regulatory regions in the genome and nuclear space, and the transcription of multiple RNA molecules. This review is a valuable summary of their series of studies. Basically, I think it is worth to be published, but I would like to make several comments.

Major comments

1) Throughout the manuscript, the phrase “mlonRNA-mediated/induced regulation (or mlonRNA-mediated chromatin conversion, etc)” is sometimes confusing because it evokes a direct action of the mlonRNA molecule itself. It should be rephrased by “mlonRNA transcription-mediated/induced xxx” in some places. The phrase “mlonRNA-mediated xxx” fits only for the later part of Section4, where the direct action of mlonRNA is mentioned.

2) There are some problems with Section 4 and I recommend the authors modify this section. The followings are what I feel are the problems in each paragraph:

- The first paragraph (LL. 259-273) is too introductory. It could be included in the Introduction section of this review.

- The second paragraph (LL. 274-292) begins with a description of lncRNAs transcribed from fbp1, but it has already been mentioned in the previous section. I think the authors should remove or minimize the detailed description of fbp1 lncRNA, and concentrate on the topic of chromatin remodeling, which is written in the second half of this paragraph.

- In the fourth paragraph, perhaps, “it can be considered that lncRNA transcription induces and/or modulates DNA supercoils” (L335-336) is what the authors want to claim in this paragraph and so this should be stated earlier in this paragraph. Before reaching this claim/topic sentence, the introductory part (the first half of this paragraph, LL. 318-334) is too long. This first half should be shortened or written as an independent paragraph.

3) L. 485: Regarding the phrase "quality control"; in general, I believe "quality control" refers to a mechanism that identifies misfolded transcripts/peptides and selectively removes them. Do the authors have an idea of how those transcription regulators work with quality control? Please discuss more the mechanism by which multiple Tup repressions contribute to quality control of the fbp1 transcript.

Minor comments

4) L. 125: “the” is italicized.

5) LL. 182: The authors described “The chromatin alteration kinetics were analyzed further in the shorter time points (10, 20, 30, and 60 min) …”, but, after this, kinetics in this shorter time points is not described. What happens in 10, 20, 30, or 60 min? If the authors add time points for each step in the text or figure, it will help the reader’s understanding.

6) LL. 183-189: In the sentence “ …and it was observed that chromatin at the fbp1 upstream region was progressively converted into an open configuration with several species of lncRNAs (three overlapping lncRNA species (-a, -b, and –c in order)) transcribed through the fbp1 upstream region [52] (Fig. 2)”, the relationship between chromatin conversion and lncRNA molecules is unclear in this sentence. The clear description here would help an easy understanding of the later claim. Although readers will later know the “transcription of mlonRNA” is a cause of the chromatin conversion, I feel it is late.

7) LL. 211-212: Regarding “… but transcription of the fbp1 gene requires Rst2.”: This sentence is confusing because the authors stated in the previous sentence “Atf1 binds to UAS1…, and also induces a cascade of mlonRNAs,” in lines 206-207, which appears that Atf1 can induce mlonRNA transcription by solely itself. The Rst2-dependency should be mentioned in the previous Atf1 part or the Rst2 part should appear earlier than the Atf1 part.

8) L. 285: the phrase “mlonRNAs transcribing RNAPII” maybe needs hyphenation like “mlonRNAs-transcribing RNAPII”.

9) L. 291: Do the “two pathways” mean Snf22 and Hrp3 pathways? It would be better to describe it clearly.

10) L. 352: Regarding “the pyp3 gene promoter region”, please provide information about pyp3: what is pyp3? Is it induced by glucose starvation? Does it have regulatory DNA elements common to fbp1 on its promoter region? And please cite the reference(s) for this part (Maybe [80]?).

11) LL. 363-364: The phrase “Although transcription of mlonRNA is important for chromatin regulation at the fbp1 promoter region” would not be necessary. It has been stated many times before. The next “transcribed mlonRNA molecules provide additional regulation involving Atf1 binding to the target locus [68]” should appear earlier.

12) LL. 366-371: Please add references for the following descriptions, “Inhibition of mlonRNA production by a transcription inhibitor or deletion of the expected mlonRNA promoter region caused a reduction in Atf1 binding in the UAS1 region. This defect in stable Atf1 binding by mlonRNA deletion was compensated by Tup11/12 deletion, indicating that this function is mediated by modulating the Tup11/12 function. RNA immunoprecipitation analysis revealed that the transcribed mlonRNA interacted with Tup11/12.”

13) Regarding Section 5, is the promoter region of the fbp1 gene a hotspot for recombination? Can the authors provide any suggestions about it in the text?

14) L. 424: Is the “initialized” correct? Perhaps the author would like to describe it as "initiated"?

15) L. 461: “Pombe” should be written in lower letters.

16) L.566 “fission fbp1” should be “fission yeast fbp1

Author Response

In this review manuscript, the authors review the regulatory mechanisms of fbp1 gene expression in fission yeast, encompassing many years of research, including their own work.

Early work by Dr. Hoffman and by Dr. Yamamoto and by the authors’ subsequent works have revealed that the fbp1 gene expression is regulated by multilayered processes, which involve multiple actions of transcriptional activators and repressors and the arrangement of transcriptional regulatory regions in the genome and nuclear space, and the transcription of multiple RNA molecules. This review is a valuable summary of their series of studies. Basically, I think it is worth to be published, but I would like to make several comments.

 (Response) Thank you for your overall positive judgment for your manuscript. We revised all issues raised bellow.

Major comments

1) Throughout the manuscript, the phrase “mlonRNA-mediated/induced regulation (or mlonRNA-mediated chromatin conversion, etc)” is sometimes confusing because it evokes a direct action of the mlonRNA molecule itself. It should be rephrased by “mlonRNA transcription-mediated/induced xxx” in some places. The phrase “mlonRNA-mediated xxx” fits only for the later part of Section4, where the direct action of mlonRNA is mentioned.

  (Response) Thank you for your comment. We agreed this opinion and changed the descriptions in this manuscript.

2) There are some problems with Section 4 and I recommend the authors modify this section. The followings are what I feel are the problems in each paragraph:

- The first paragraph (LL. 259-273) is too introductory. It could be included in the Introduction section of this review.

(Response) Thank you for your comment. We agreed this opinion and moved this paragraph to introduction section.

- The second paragraph (LL. 274-292) begins with a description of lncRNAs transcribed from fbp1, but it has already been mentioned in the previous section. I think the authors should remove or minimize the detailed description of fbp1 lncRNA, and concentrate on the topic of chromatin remodeling, which is written in the second half of this paragraph.

(Response) Thank you for your comment. We agreed this opinion and removed most of the first half of this paragraph and concentrated on the topic of chromatin remodeling associating with mlonRNA-transcription. With this modification, we also removed description about fbp1-as lncRNA from Fig. 3A, which seems to be not a main topic of this review focusing on the chromatin modification at fbp1 locus.

- In the fourth paragraph, perhaps, “it can be considered that lncRNA transcription induces and/or modulates DNA supercoils” (L335-336) is what the authors want to claim in this paragraph and so this should be stated earlier in this paragraph. Before reaching this claim/topic sentence, the introductory part (the first half of this paragraph, LL. 318-334) is too long. This first half should be shortened or written as an independent paragraph.

(Response) Thank you for your comment. We agreed this opinion and shortened the first half of this paragraph.

3) L. 485: Regarding the phrase "quality control"; in general, I believe "quality control" refers to a mechanism that identifies misfolded transcripts/peptides and selectively removes them. Do the authors have an idea of how those transcription regulators work with quality control? Please discuss more the mechanism by which multiple Tup repressions contribute to quality control of the fbp1 transcript.

(Response) Thank you for your comment. We agreed this opinion and omitted the used of the term ‘quality control’.

Minor comments

4) L. 125: “the” is italicized.

(Response) Thanks. We fixed this error.

5) LL. 182: The authors described “The chromatin alteration kinetics were analyzed further in the shorter time points (10, 20, 30, and 60 min) …”, but, after this, kinetics in this shorter time points is not described. What happens in 10, 20, 30, or 60 min? If the authors add time points for each step in the text or figure, it will help the reader’s understanding.

(Response) Thanks. With the next comment, we edited this description to improve the suggested points. (Line 223-233)

6) LL. 183-189: In the sentence “ …and it was observed that chromatin at the fbp1 upstream region was progressively converted into an open configuration with several species of lncRNAs (three overlapping lncRNA species (-a, -b, and –c in order)) transcribed through the fbp1 upstream region [52] (Fig. 2)”, the relationship between chromatin conversion and lncRNA molecules is unclear in this sentence. The clear description here would help an easy understanding of the later claim. Although readers will later know the “transcription of mlonRNA” is a cause of the chromatin conversion, I feel it is late.

(Response) Thanks. We modified the description here as suggested. (Line 223-233)

7) LL. 211-212: Regarding “… but transcription of the fbp1 gene requires Rst2.”: This sentence is confusing because the authors stated in the previous sentence “Atf1 binds to UAS1…, and also induces a cascade of mlonRNAs,” in lines 206-207, which appears that Atf1 can induce mlonRNA transcription by solely itself. The Rst2-dependency should be mentioned in the previous Atf1 part or the Rst2 part should appear earlier than the Atf1 part.

 (Response) Thank you for your comment. We agreed this opinion and changed this sentence as follows.

“but transcription of the fbp1-mRNA from TATA-box still requires Rst2” 

8) L. 285: the phrase “mlonRNAs transcribing RNAPII” maybe needs hyphenation like “mlonRNAs-transcribing RNAPII”.

(Response) Thanks. We fixed this error.

9) L. 291: Do the “two pathways” mean Snf22 and Hrp3 pathways? It would be better to describe it clearly.

(Response) Thanks. We fixed this.

10) L. 352: Regarding “the pyp3 gene promoter region”, please provide information about pyp3: what is pyp3? Is it induced by glucose starvation? Does it have regulatory DNA elements common to fbp1 on its promoter region? And please cite the reference(s) for this part (Maybe [80]?).

(Response) Thanks. We added following description.

“ , which promoter is constitutively active without requirement of the glucose starvation signal.”

11) LL. 363-364: The phrase “Although transcription of mlonRNA is important for chromatin regulation at the fbp1 promoter region” would not be necessary. It has been stated many times before. The next “transcribed mlonRNA molecules provide additional regulation involving Atf1 binding to the target locus [68]” should appear earlier.

  (Response) Thank you for your comment. We agreed this opinion and removed this sentence.

12) LL. 366-371: Please add references for the following descriptions, “Inhibition of mlonRNA production by a transcription inhibitor or deletion of the expected mlonRNA promoter region caused a reduction in Atf1 binding in the UAS1 region. This defect in stable Atf1 binding by mlonRNA deletion was compensated by Tup11/12 deletion, indicating that this function is mediated by modulating the Tup11/12 function. RNA immunoprecipitation analysis revealed that the transcribed mlonRNA interacted with Tup11/12.”

   (Response) Thank you for your comment. We added the citation here.

13) Regarding Section 5, is the promoter region of the fbp1 gene a hotspot for recombination? Can the authors provide any suggestions about it in the text?

   (Response) Thank you for your comment. We are currently analyzing this possibility. The fbp1 is not meiotic recombination hotspot, but if cells suffer from glucose during meiosis, this region become meiotic recombination hotspot (unpublished data). Since this is unpublished result, we would like to avoid the statement here and provide data to reader in near future elsewhere.

14) L. 424: Is the “initialized” correct? Perhaps the author would like to describe it as "initiated"?

 (Response) Thanks. We fixed this error.

15) L. 461: “Pombe” should be written in lower letters.

 (Response) Thanks. We fixed this error.

16) L.566 “fission fbp1” should be “fission yeast fbp1

 (Response) Thanks. We fixed this error.